# Effect of Temperature and Radiation on *Indica* Rice Yield and Quality in Middle Rice Cropping System

**DOI:** 10.3390/plants11202697

**Published:** 2022-10-13

**Authors:** Debao Tu, Wenge Wu, Min Xi, Yongjin Zhou, Youzun Xu, Jinhua Chen, Caihong Shao, Yuping Zhang, Quanzhi Zhao

**Affiliations:** 1Rice Research Institute, Anhui Academy of Agricultural Sciences, Hefei 230031, China; 2Anhui Agrometeorological Institute, Hefei 230031, China; 3Red Soil Engineering and Technology Center, Soil and Fertilizer & Resource and Environment Institute, Jiangxi Academy of Agricultural Sciences, Nanchang 330200, China; 4China National Rice Research Institute, Chinese Academy of Agricultural Sciences, Hangzhou 310006, China; 5College of Agronomy, Henan Key Laboratory of Regulation and Control of Crop Growth and Development, Henan Agricultural University, Zhengzhou 450002, China

**Keywords:** rice, temperature, radiation, yield, quality

## Abstract

Rice (*Oryza sativa* L.) is cultivated in a wide range of climatic conditions, thereby inducing great variations in the rice growth, yield and quality. However, the comprehensive effects of temperature and solar radiation under different ecological regions on the rice growth, yield and quality are not well understood, especially in a middle rice cropping system. The rice growth, yield- and quality-related traits were investigated under different ecological regions. Among different areas, the days before the heading stage and after the heading stage of six cultivars ranged from 80 to 120 and from 30 to 50. The gaps of the grain yield, head rice rate, chalky grain rate and chalkiness level were about 1.2–52.4%, 1.0–3.0%, 2.7–12.7% and 0.3–4.5%, respectively. This study demonstrated that in these regions, temperature is a limiting factor compared with radiation. Moreover, the rice growth, yield and quality were closely associated with daily air (DT), maximum (MaT), minimum (MiT) and effective accumulated temperatures (EAT). An excellent rice growth, a high grain yield and an excellent quality could be achieved if the EAT was higher than 1592 °C·d and the MiT was lower than 23.1 °C before the heading stage, and if the DT, MiT and MaT were lower than 25.7 °C, 22.0 °C and 30 °C after the heading stage, respectively. These findings served as an important reference for optimizing cultivar selection for a specific area and determining suitable areas for a certain variety.

## 1. Introduction

Rice (*Oryza sativa* L.) provides a fulfilling number of calories and nutritional content for more than half of the population around the world [1]. The global demand for rice is expected to increase by 28% in the next three decades [2]. In addition, the human demand for rice quality is enhancing since the living standards of society and the economy have markedly improved over the course of the past decades [3,4]. However, there are great challenges for improving the grain yield and quality due to climate change [5,6]. To meet the challenges, the climatic conditions that affect the grain yield and quality should be understood and managed better for less degradation and input and more output.

Numerous studies investigating the influence of temperature on grain development have indicated that high temperatures significantly decreased the grain yield and shortened the rice growth duration [5,7]. For the rice quality, a high temperature would result in a high chalky grain rate, high chalkiness level [8,9,10], low head rice rate [11,12] and the deterioration of the grain’s physicochemical [6,13]. Additionally, several studies have stated that solar radiation is closely associated with photosynthesis and plays a central role in determining the rice grain yield and quality [14,15,16]. Hence, both the temperature and solar radiation play key roles in determining rice grain yield as well as grain quality.

China, the largest rice producer, plays an important role in meeting food security. The middle rice cropping system is the leading cultivation, accounting for about 50% [17]. In the past decades, the large areas of the double rice cropping system were converted to middle rice cropping in east and central China [18,19], while the area of the middle rice cropping system is increasing and will be cultivated in a wide range of areas in east and central China [17]. The rice crop in each location is subjected to different sets of temperature and solar radiation, and these are differences that expect to be amplified with climate change. Therefore, it is important to understand the grain yield and quality response of different *indica* rice varieties to temperature and solar radiation, and thus suitable rice varieties can then be recommended based on their temperature and solar radiation requirements in order to achieve good qualities and a high grain yield.

Although previous studies have established that there are differences in yield traits in rice under different temperature and radiation conditions [7], the comprehensive effects of the natural field temperature and solar radiation under different ecological regions are not well understood. Furthermore, most previous research examined the effects of climatic conditions only on the grain yield or quality [3,5,6]. Therefore, field experiments were conducted to observe the impact of climatic factors under four ecological regions with different *indica* rice cultivars, the main type of rice in east and central China. The objectives of the study are: (1) to explore how the different quality characteristics and grain yield of *indica* rice are affected when grown under different ecological regions and (2) to assess the correlation of climatic conditions with the grain yield and quality.

## 2. Results

### 2.1. Differences of Temperature and Solar Radiation Conditions across Four Areas and Two Years

Before the heading stage, the EAT (effective accumulated temperature) showed a slight difference among different ecological regions, except in Yifeng (Table 1). The EAT of six cultivars in both years showed the same pattern under each area, except in Yuyang. However, the DT, MaT and TD showed significant differences among different areas and years. The daily solar radiation and the CSR were higher in 2019 than in 2020. In addition, the CSR in Yuanyang and Tongcheng was significantly higher than that in Fuyang and Yifeng in 2019, while there were slight differences among different areas in 2020, except in Yuanyang. The daily solar radiation also demonstrated large variations among different areas and between the two years. After the heading stage, there were significant differences in the EAT, DT, MaT, MiT and TD across the different areas, as well as there were some differences between the two years. Similarly, the solar radiation and CSR demonstrated differences among the areas, but there were no differences between the two years, except in Tongcheng and Yifeng.

### 2.2. The Variation in Rice Growth Duration across Four Areas and Two Years

The rice growth durations were also examined and showed in Figure 1. The days before the heading stage (SD-HD) of the six cultivars were from 80 to 120 across different areas. It was higher in Tongcheng (31.0° N), followed by Fuyang (30.0° N), Yuangyang (35.3° N) and Yifeng (28.4° N). It showed the same tendency in 2019 and 2020. Meanwhile, there were also some differences in the days before the heading stage between the two years. For the days after the heading stage (HD-MD), it was from 30 to 50 for six cultivars across the different areas. It was the highest in Tongcheng (31.0° N) and the lowest in Fuyang (30.0° N). Similarly, there were also some differences in the days after the heading stage between the two years.

### 2.3. The Difference of Grain Yield and Yield Components across Four Areas and Two Years

In 2019, the average yield of the six cultivars was the lowest in Yifeng (28.4° N), with 52.3%, 28.6% and 31.7% lower in Yuangyang (35.3° N), Fuyang (30.0° N) and Tongcheng (31.0° N), respectively (Figure 2). In 2020, the results showed that the yield in Yifeng (28.4° N) was 1.2%, 18.5% and 28.4% lower in Yuangyang (35.3° N), Fuyang (30.0° N) and Tongcheng (31.0° N), respectively. In addition, the yield in each area in 2020 was higher than the yield in 2019, except in Yuangyang (35.3° N). Although this tendency was not always significant in some cultivars, the overall tendencies of the yields of the six cultivars were identical across the different areas each year.

A correlation analysis between the grain yield and yield components was performed in Figure 3. The relative grain yield (RGY) demonstrated a greater correlation with the relative panicle m^−2^ (RP, r = 0.676, *p* < 0.001) and the relative grain filling percentage (RGP, r = 0.316, *p* = 0.041) than with the relative spikelets panicle^−1^ (RS, r = 0.242, *p* = 0.122) and the relative 1000 grain weight (RGW, r = −0.048, *p* = 0.762).

### 2.4. The Difference of Grain Quality across Four Areas and Two Years

The head rice rate, chalky grain rate, chalkiness level, grain length and grain width of the six cultivars across the four areas and two years are showed by the box plot graph (Figure 4). The heading rate was lowest in Yifeng (28.4° N) in 2019, and it increased by 2.8%, 2.5% and 3.0% in Yuanyang (35.3° N), Fuyang (30.0° N) and Tongcheng (31.0° N), respectively. While it was lowest in Fuyang (30.0° N) in 2020, and the gaps of it ranged from 1.0% to 2.3% among the different areas. There were significant differences in the head rice rate among the different areas, and even between the two years. Conversely, the chalky rice rate and chalkiness level were highest in Yifeng (28.4° N) in 2019 and in Fuyang (30.0° N) in 2020. The gaps of the chalky grain rate and chalkiness level across the four areas in the two years were from 2.7% to 12.7% and from 0.3% to 4.5%, respectively. For the grain shape, both years showed the same tendency that the grain length and grain width were higher in Yuangyang (35.3° N) followed by Tongcheng (31.0° N), Fuyang (30.0° N) and Yifeng (28.4° N).

### 2.5. Correlation Analysis of Rice Growth Duration, Grain Yield- and Quality-, with Temperature

A correlation analysis was performed to determine the effect of the temperature and solar radiation on the growth duration, grain yield and quality (Table 2). The relative SD-HD (RHD-SD) showed a significantly negative correlation with the DT (daily air temperature), MaT (maximum temperature) and MiT (minimum temperature) before the heading stage. Similarly, the relative HD-MD (RHD-MD) demonstrated a markedly negative correlation with the DT, MaT and MiT after the heading stage. Hence, the relative head rice rate (RHR) and relative grain length (RL) demonstrated significantly negative correlations with the MiT before the heading stage and with the DT, MaT and MiT after the heading stage, respectively. The relative grain width (RW) showed significantly negative correlations with the MiT. While the relative chalky grain rate (RCR) and the relative chalkiness level (RCL) markedly increased with the DT and MaT after the heading stage, the RCR also positively correlated with the MiT before the heading stage.

For the grain yield, the relative grain yield (RGY) was negatively associated with the DT and MaT after the heading stage. The relative panicle m^−2^ was also negatively related to the DT and MaT after the heading stage. In addition, the relative spikelets panicle (RS) was negatively related to the DT and MaT before the heading stage. The relative grain filling percentage (RGP) negatively correlated with the MaT after the heading stage. Regardless of the mean temperature, the EAT showed a significant correlation with the growth duration, grain yield and quality. The EAT before the heading stage positively related to the RSD-HD, RSD-MD, RGY and RP (relative panicle m^−2^). However, the RGY, RS and RGP were greatly negative related to the TD after the heading stage and the RGY was significant negative associated with the EAT after the heading stage, which went against the common sense that the high TD and EAT were beneficial to the grain filling and yield formation. The reason for this result was that a high DT and MaT tended to be accompanied by a higher EAT and TD, and the former have significantly negative influences on grain yield.

### 2.6. Optimal Temperature Range for High Yield and Quality

Compared with the temperature, the solar radiation also demonstrated a great correlation with the growth duration, grain yield and quality. However, the growth duration, grain yield and quality were negatively associated with the solar radiation, which went against the common sense of the positive effect of solar radiation on the rice growth, grain yield and quality. This result may be attributed to the fact that there were confounding effects of temperature and solar radiation in these areas, which means that a high temperature tended to be accompanied by a higher solar radiation, and the former have significantly negative influences on the rice growth duration, grain yield and quality. The correlation analysis revealed that the temperature was the main factor that affected the growth duration, grain yield and quality under the different areas.

Furthermore, the result also showed that the RGY was higher than one, when the EAT before the heading stage was higher than 1592 °C·d, and the DT and MaT after the heading stage were lower than 27.2 °C and 30 °C, respectively (Figure 5). For the rice quality, the RHR was greater than 1 if the MiT before the heading stage, and the DT, MaT and MiT after the heading stage, were lower than 23.1 °C, 27.2 °C, 32.0 °C and 22.0 °C, respectively. For the RCR, it was less than 1 when the MiT before the heading stage, and the DT and MaT after the heading stage, were below 23.1 °C, 25.7 °C and 30.0 °C, respectively. Similarly, if the DT and MaT after the heading stage were lower than 25.7 °C and 30.0 °C, the RCL could be also less than 1.

## 3. Discussion

### 3.1. Effect of Climate Conditions on Rice Growth, Yield and Quality under Different Cultivated Areas

Besides the effect controlled by the genetic factors, the agronomic traits (the grain yield, quality and growth duration) are also greatly affected by the climate conditions [3,20]. For example, the rice quality can vary inexplicably from the different areas and different years [3,21]. Similarly, the results also demonstrated that there were significant differences in the grain yield, head rice rate, chalkiness and grain shape among the different areas and years in the present study. The reason for these results may be attributed by the variations in the growth duration among the different cultivated areas and years. As previous studies have proved, rice growth duration determines the total amount of incident solar radiation, and a high interception of solar radiation by the rice canopy is closely associated with a high biomass production and the grain yield [22,23]. A shortened grain filling duration would result in a low head rice rate and a high chalky grain rate [21]. These results suggested that the grain yield and quality of the same variety would change greatly due to the variations in the growth duration among the different areas.

The temperature is one of the most important climate conditions influencing the rice growth and development, and the final formation of the grain yield and quality [6,24]. Likewise, the RSD-HD demonstrated significantly negative correlations with the DT, MaT and MiT before the heading stage, as well as it negatively correlated with the DT, MaT and MiT after the heading stage. It was consistent with the previous reports that a high temperature during the vegetative stage accelerated the rice growth rate, leading to heading and flowering 4–7 days earlier than usual [25], and a high temperature also accelerated the leaf senescence and shortened the duration of the grain filling [26]. The shortened growth duration would decrease the EAT, biomass production, panicle per m^−2^, spikelets panicle^−1^, grain filling percentage and grain yield. In addition, a high temperature also induced small caryopsis, inhibited cell development and caused starch deposition, which plays an important role in the formation of chalkiness [1,8,27]. Similarly, the results showed that the RL and RW demonstrated a significantly negative correlation with the MiT before the heading stage, and the DT, MaT and MiT after the heading stage. These eventually resulted in a high chalky grain rate and chalkiness level and a low head rice rate. As previous studies have suggested, high temperature induced a high chalkiness and a low head rice rate [6,9]. Obviously, there were great differences in the DT, MiT and MaT before the heading stage and after the heading stage of *indica* rice among different areas in the present study. Therefore, we suggested that temperature, namely the DT, MiT, MaT and EAT, played a key role in determining the rice growth and the final formation of grain yield and quality among the different areas.

Indeed, temperature is related to other meteorological factors such as solar radiation, humidity and light hours [21]. Much evidence demonstrated that the radiation had a positive impact on the rice growth and development [14,15]. Shading significantly increased the chalky grain rate and reduced the head rice rate [16]. However, compared with the temperature, solar radiation showed a slighter correlation with the grain yield and quality. It was consistent with the finding that the temperature is the most important climatic factor governing the growth duration of rice [21,28]. Moreover, solar radiation demonstrated strongly negative correlations with the grain yield and quality in the present study, which went against the common sense of the positive influence of radiation on the grain yield and quality [29,30]. This result may be attributed to the fact that there were confounding effects of temperature and solar radiation [7]. This means that a high temperature always accompanied high solar radiation in these regions, and the former has significantly negative effects on the grain yield and quality [8,10]. Similarly, previous studies have shown that solar radiation is not a limiting factor to the grain yield and quality, compared with the temperature in these regions [5,7]. Therefore, we suggested that the variations of grain yield and quality among the different areas were mainly attributed to the temperature changes.

### 3.2. Optimal Climate Conditions Range for Achieving High Grain Yield and Quality

Obviously, the correlation analysis results suggested that the grain yield was determined by the EAT before the heading stage, and the DT and MaT after the heading stage. For the rice quality, the head rice rate was closely associated with the MiT before the heading stage, and the DT, MaT and MiT after the heading stage. Previous studies also suggested that DTs of 26–28 °C during the vegetative stage and 22–27 °C during the grain filling stage are required to attain a high grain yield and quality [7]. According to the definition of the relative data, the GY, HR, CR and CL were higher than average for a certain variety if the RGY, RHR, RCR and RCL were greater than one, respectively [5,6]. Our results suggested that the EAT was higher than 1592 °C·d, while the MiT was lower than 23.1 °C before the heading stage, as well as the DT, MiT and MaT were lower than 25.7 °C, 22.0 °C and 30 °C after the heading stage; this was helpful in achieving a high grain yield and quality in the middle rice cropping system. 

Furthermore, there were also some differences in the rice growth duration, yield and quality among the different cultivars (Figure 1, Figure 2 and Figure 4), which is closely associated with different temperature sensitivities for the cultivars [7,8]. For example, our results also showed that a high yield and high quality may have been achieved sometimes when the temperature values were out of the optimal temperature. This result suggested we could also obtain a high grain yield and quality by selecting high heat-resistant varieties [8]. Despite the fact that the cultivars contained different temperature sensitivities, the overall tendencies of the rice growth, yield and quality were identically associated with the temperature (EAT, DT, MiT, and MaT) in the present study. Thus, the findings in this study served as an important reference for optimizing cultivar selection for a specific area and determining the suitable planting areas for a certain variety, in order to achieve a high grain yield and quality.

## 4. Materials and Methods

### 4.1. Site Description and Experimental Design

The field experiments were conducted in 2019 and repeated in 2020, under different ecological regions including Yuanyang in Henan province, China (35.3° N, 114.1° E), Tongcheng in Anhui province, China (31.0° N, 116.9° E), Fuyang in Zhejiang province, China (30.0° N, 119.9° E) and Yifeng in Jiangxi province, China (28.4° N, 114.8° E). The properties of the soil determined from the upper 20 cm layer were showed in Appendix A: it had a pH value of 5.9–6.8, an organic matter of 15.6 g kg^−1^–31.1 g kg^−1^, a total N of 1.4 g kg^−1^–2.5 g kg^−1^, an available Olsen-P of 11.1 mg kg^−1^–26.5 mg kg^−1^ and an exchangeable K of 66.3 mg kg^−1^–109.8 mg kg^−1^. In the present study, six *indica* rice cultivars with different temperature sensitivities were established for each region, namely Taoyouxiangzhan, Taiyou398, Fengliangyou4hao, Jingliangyouhuazhan, Wandao153 and IIyou838. They were sown in May and headed in August (Appendix A), which is the typical middle rice systems’ rice growth season. The experiments were arranged in a randomized complete block design with three replications. The seedlings were transplanted at a hill spacing of 30 cm × 15 cm.

The total N, P and K fertilizers were applied at the amount of 195, 70 and 135 kg hm^−2^, respectively. The fertilizers were used in the forms of a compound fertilizer, urea, calcium superphosphate and potassium chloride. Fertilizer-N was applied in three splits: 55% as basal, 20% at tillering and 25% one week after the panicle initiation. Fertilizer-P was fully applied as basal, while fertilizer-K was applied in two splits of 50% as basal and 50% one week after the panicle initiation. In the present study, the experimental fields have a good irrigation system. The field was flooded after transplanting, and a floodwater depth of 3–5 cm was maintained until a week before maturity, except that the water was drained at the maximum tillering stage to reduce the unproductive tillers. Insects, diseases and weeds were intensively controlled by chemicals to avoid biomass and grain yield losses.

### 4.2. Meteorological Data Collection

The meteorological data in four regions from 2019 to 2020 were derived from the local weather station AWS 800 (Campbell Scientific, Campbell, CS, USA) located near the experimental field. The data included the daily air temperature (DT), maximum temperature (MaT), minimum temperature (MiT), light hours and humidity. The humidity, light hours and precipitation conditions during rice growth seasons across four cultivated areas and two years were showed in Appendix A. In addition, the solar radiation data were simulated based on the light hours according to the empirical model of the Angstrom function [5,7]:(1)$$\frac{R_{G}}{R_{A}}=a+b\times \frac{n}{N}$$
where *R_G_* is the global solar radiation (MJ m^−2^ day^−1^), *R_A_* is the extraterrestrial solar radiation (MJ m^−2^ day^−1^), *n* is the actual light hours in a day and *N* is the potential light hours in a day. According to the method of He and Xie [31], the empirical coefficients “a” and “b” were 0.126 and 0.648, respectively.

### 4.3. Grain Yield and Yield Components

At the maturity stage, plants from an area of 5 m^2^ in the center of the plot were harvested and, subsequently, the grain yield was adjusted to 13.5% moisture content. Twelve plant samples from each plot were collected randomly (with the three outermost rows removed to minimize the border effect) to calculate the yield components. Panicles were hand-threshed and filled and the unfilled grains were separated by submerging them in tap water; a seed blower (CFY-II, China) was used to separate the half-filled and empty grains. Subsamples were taken to manually count the total number of filled, half-filled and empty grains to assess the number of grains per panicle and the seed setting rate. In addition, the grain weight was estimated from the filled grains. Grain yield of each cultivar across four areas and two years was showed in Appendix A.

### 4.4. Determination of Rice Quality

The mature rice was threshed after harvest, air-dried and stored at room temperature for 3 months until testing. A rubber roller sheller (BLH-3250, Zhejiang Bethlehem Aparatus Co., Ltd., Zhejiang, China) was used to shell them first. Then, the brown rice was milled with a rice polishing machine (Kett, Tokyo, Japan) and the head rice rate was measured. In addition, the appearance quality of the rice was measured and analyzed with a scanner Epson Expression 1680 Professional (Epson, Nagano-ken, Japan) and using image analysis software. Grain quality of each cultivar across four areas and two years was showed in Appendix A.

### 4.5. Calculation Methods and Statistical Analysis

The average daily climatic data from sowing to heading (before heading) and from heading to maturity (after heading) were calculated. In order to identify the response of rice growth to the temperature and solar radiation change, the differences in the properties among the cultivars were minimized by using the average and standardized data management [5,6]. The standardized data were defined as “relative data” and calculated as follows:Relative data = (observed data of a variety during at a specific study site in one years)/(mean data of this variety across study sites in both years)

The temperature different (TD), effective accumulated temperature (EAT) and cumulative solar radiation (CSR) were calculated as follows:TD (°C) = MaT − MiT(2)
EAT (°C) = ∑(MT − T_0_) × growth duration (GD)(3)
CSR (MJ m^−2^) = ∑(R) × growth duration (GD)(4)
where T_0_ (10 °C for rice) is the biological zero temperature [6] and R is the daily solar radiation. 

A one-way analysis of variance (ANOVA) was used for the analysis of the grain yield and quality variations in this study. The Pearson correlation analysis was conducted to determine the relationship among the variations of climatic data and the relevant agronomic traits by using SPSS. In addition, the graphs were prepared by using SigmaPlot 14.0.

## 5. Conclusions

The temperature is the main climatic factor influencing the rice growth, yield and quality in China. It is necessary to understand the effects of temperature on the rice growth, yield potential and quality, clearly. This study indicated that in these regions, the temperature is a limiting factor compared with radiation. In the present study, the results revealed that the rice growth, yield and quality of six rice varieties showed significantly positive correlations with the daily air (DT), maximum (MaT), minimum (MiT) and effective accumulated temperatures (EAT). Furthermore, our result suggested that an excellent rice growth, a high grain yield and an excellent quality could be achieved if the EAT was higher than 1592 °C·d and the MiT was lower than 23.1 °C before the heading stage. Additionally, great performances could be attained when the DT, MiT and MaT were lower than 25.7 °C, 22.0 °C and 30 °C after the heading stage, respectively. These findings provide important references for production management to obtain a high grain yield and quality.

## Figures and Tables

**Figure 1 plants-11-02697-f001:**
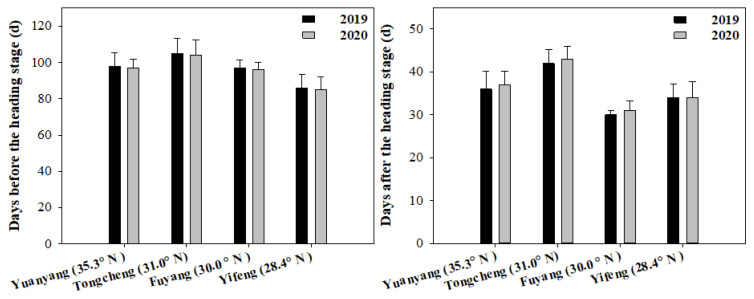
The rice growth duration before the heading and after heading stages in different areas in 2019 and 2020.Vertical bars represent ± standard error of the means of six cultivars (*n* = 6).

**Figure 2 plants-11-02697-f002:**
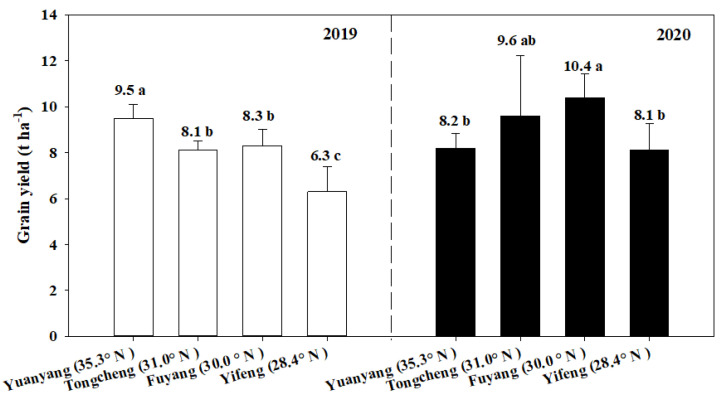
The grain yield in different areas in two years. The value represents the mean yield of six cultivars in each area and year. Vertical bars represent ± standard error of the means of six cultivars (*n* = 6). Means followed by different letters are significantly different according to LSD (0.05) for each year.

**Figure 3 plants-11-02697-f003:**
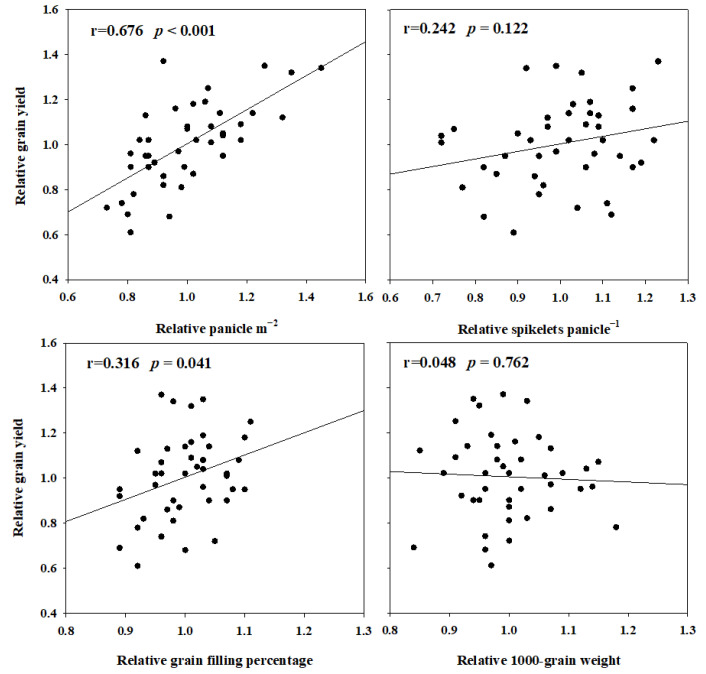
The relationship of the relative grain yield (RGY) with relative panicle m^−2^ (RP), relative spikelets panicle^−1^ (RS), relative grain filling percentage (RGP) and relative 1000-grain weight. Data were pooled from experiments conducted in four cultivated areas of 2019 and 2020.

**Figure 4 plants-11-02697-f004:**
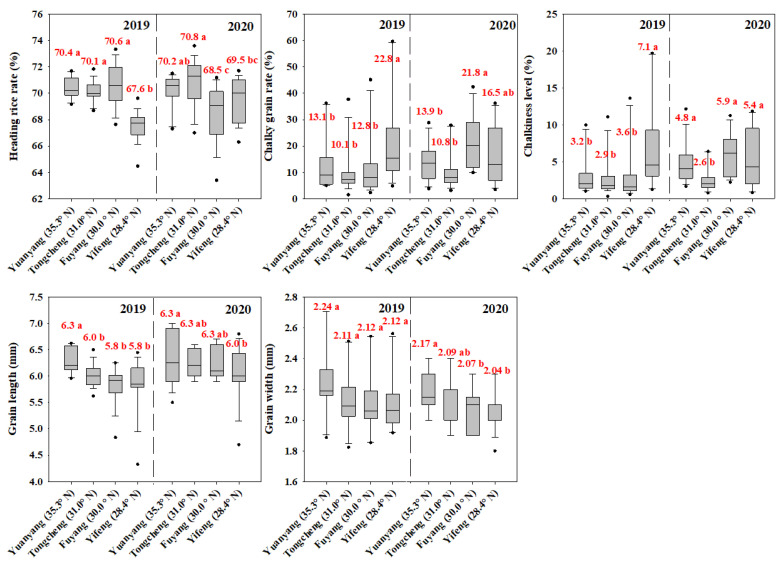
The box plot graph of the rice quality traits (including head rice rate, chalky grain rate, chalkiness level, grain length and grain width) of six cultivars in different areas. The value represents the mean yield of six cultivars in each area and year. Means followed by different letters are significantly different according to LSD (0.05) for each year.

**Figure 5 plants-11-02697-f005:**
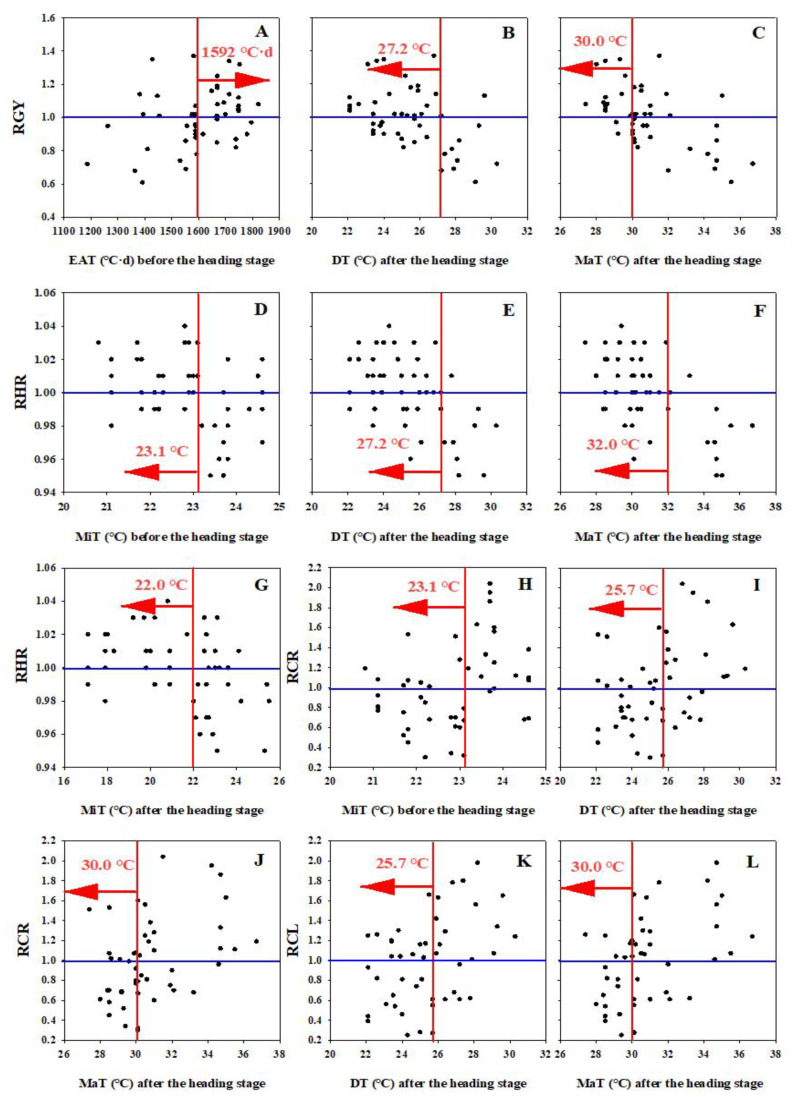
Grain yield (**A**–**C**), head rice rate (**D**–**G**), chalky grain rate (**H**–**J**) and chalkiness level (**K**,**L**) trends with temperature condition. The blue line represents temperatures for RGY, RHR, RCR and RCL at 1.0. The red line is the temperature threshold dividing line.

**Table 1 plants-11-02697-t001:** Temperature and solar radiation condition before heading stage and after heading stage among different areas in 2019 and 2020.

Year	Duration	Area	EAT ^a^ (°C·d)	DT_mean_ (°C)	MaT_mean_ (°C)	MiT_mean_ (°C)	TD_mean_ (°C)	CSR (MJ m^−2^)	R_mean_(MJ m^−2^ d^−1^)
2019	Before heading stage	Yuanyang	1682 ± 128.1 a ^b^	27.2 ± 0.09 b	33.1 ± 0.18 a	21.7 ± 0.03 f	11.4 ± 0.21 a	1838 ± 124.3 a	18.7 ± 0.21 a
		Tongcheng	1689 ± 154.4 a	26.0 ± 0.25 e	30.8 ± 0.29 e	22.1 ± 0.24 e	8.6 ± 0.05 c	1896 ± 163.3 a	18.0 ± 0.20 b
		Fuyang	1606 ± 83.7 ab	26.6 ± 0.13 d	31.0 ± 0.16 de	23.1 ± 0.09 c	8.0 ± 0.08 d	1501 ± 86.2 bc	15.6 ± 0.22 d
		Yifeng	1468 ± 154.2 b	27.1 ± 0.32 bc	32.2 ± 0.45 b	23.6 ± 0.18 b	8.6 ± 0.28 c	1535 ± 193.7 b	17.8 ± 0.75 b
2020		Yuanyang	1557 ± 79.4 ab	26.0 ± 0.01 e	31.4 ± 0.07 c	21.0 ± 0.11 g	10.3 ± 0.18 b	1592 ± 53.5 b	16.4 ± 0.30 c
		Tongcheng	1676 ± 159.9 a	26.1 ± 0.29 e	30.3 ± 0.28 f	22.7 ± 0.30 d	7.5 ± 0.04 e	1353 ± 131.5 cd	13.0 ± 0.25 e
		Fuyang	1613 ± 88.7 ab	26.9 ± 0.24 c	31.1 ± 0.28 cd	23.7 ± 0.16 b	7.4 ± 0.12 e	1172 ± 108.2 e	12.3 ± 0.66 f
		Yifeng	1499 ± 136.1 b	27.6 ± 0.17 a	32.4 ± 0.20 b	24.5 ± 0.13 a	7.8 ± 0.07 d	1299 ± 136.2 de	15.3 ± 0.41 d
2019	After the heading stage	Yuanyang	457 ± 81.9 c	22.5 ± 0.74 e	28.6 ± 0.31 c	17.7 ± 1.03 c	10.9 ± 0.74 b	586 ± 67.2 c	16.1 ± 0.44 bc
		Tongcheng	633 ± 78.9 a	25.0 ± 1.08 cd	30.1 ± 1.01 bc	20.9 ± 1.17 b	9.2 ± 0.22 c	667 ± 74.9 b	15.8 ± 0.97 bcd
		Fuyang	485 ± 19.2 bc	26.2 ± 0.63 bc	30.7 ± 0.81 b	22.8 ± 0.46 a	7.9 ± 0.35 d	436 ± 30.9 d	14.5 ± 1.18 cd
		Yifeng	623 ± 28.9 a	28.5 ± 1.04 a	35.1 ± 0.91 a	23.4 ± 1.25 a	11.7 ± 0.37 a	742 ± 42.6 a	22.0 ± 0.99 a
2020		Yuanyang	512 ± 59.9 bc	23.7 ± 0.50 de	30.2 ± 0.33 bc	18.3 ± 0.70 c	11.9 ± 0.48 a	648 ± 37.5 bc	17.4 ± 0.60 b
		Tongcheng	606 ± 98.6 a	24.1 ± 1.60 d	29.0 ± 1.60 c	20.6 ± 1.56 b	8.3 ± 0.15 d	590 ± 71.9 c	13.7 ± 1.09 d
		Fuyang	499 ± 29.3 bc	26.5 ± 1.64 b	31.2 ± 1.97 b	23.0 ± 1.18 a	8.2 ± 0.79 d	446 ± 76.4 d	14.8 ± 3.51 cd
		Yifeng	561 ± 22.1 ab	26.6 ± 1.71 b	31.5 ± 2.09 b	23.1 ± 1.35 a	8.4 ± 0.76 d	601 ± 38.3 bc	17.8 ± 2.71 b

^a^ T_mean_, MaT_mean_, MiT_mean_, TD_mean_ and R_mean_ represent the mean daily air temperature, maximum temperature, minimum temperature, temperature difference and solar radiation of six cultivars (±S.E.) before heading stage and after heading stage, respectively. EAT is the effective accumulated temperature. CSR is the cumulative solar radiation. ^b^ Means followed by different letters are significantly different according to LSD (0.05) for each year and each growth stage. Different ecological regions including Yuanyang in Henan province (35.3° N, 114.1° E), Tongcheng in Anhui province (31.0° N, 116.9° E), Fuyang in Zhejiang province (30.0° N, 119.9° E) and Yifeng in Jiangxi province (28.4° N, 114.8° E). Statistical analysis was performed by one-way ANOVA.

**Table 2 plants-11-02697-t002:** Correlation of growth duration, grain yield- and quality-related traits with temperature and solar radiation before heading stage and after heading stage.

	Before Heading Stage								After Heading Stage						
	DT ^a^	EAT	MaT	MiT	TD	CSR	R		DT	EAT	MaT	MiT	TD	CSR	R
RSD-HD ^b^	−0.717 **	0.585 **	−0.559 **	−0.591 **	0.082	0.389 **	−0.067		−0.582 **	−0.031	−0.635 **	−0.481 **	−0.157	−0.168	−0.595 **
RHD-MD	−0.440 **	0.197	−0.240	−0.415 **	0.171	0.333 *	0.190		−0.424 **	0.623 **	−0.403 **	−0.391 **	0.021	0.537 **	−0.249
RHR	−0.267	0.229	−0.170	−0.373 **	0.173	0.185	0.028		−0.504 **	−0.118	−0.586 **	−0.366 *	−0.259	−0.205	−0.520 **
RL	−0.130	0.167	0.011	−0.309 *	0.249	−0.005	−0.149		−0.363 *	−0.196	−0.299 *	−0.372 **	0.137	−0.060	−0.166
RW	−0.128	0.092	0.165	−0.417 **	0.454 **	0.323 *	0.315 *		−0.272	−0.150	−0.126	−0.373 **	0.364 *	0.122	0.062
RCR	0.261	−0.148	0.140	0.323 *	−0.154	−0.272	−0.167		0.345 *	0.030	0.385 **	0.265	0.140	0.043	0.277
RCL	0.220	−0.240	0.132	0.284	−0.126	−0.271	−0.130		0.386 **	0.077	0.465 **	0.261	0.250	0.161	0.405 **
RGY	−0.031	0.394 **	−0.129	−0.078	−0.025	−0.138	−0.403 **		−0.457 **	−0.333 *	−0.590 **	−0.275	−0.387 **	−0.503 **	−0.636 **
RP	0.026	0.342 *	−0.037	−0.026	−0.007	0.051	−0.129		−0.432 **	−0.063	−0.570 **	−0.230	−0.474 **	−0.287	−0.559 **
RS	−0.353 *	0.051	−0.354 *	−0.213	−0.070	−0.163	−0.298		−0.030	−0.130	0.000	−0.078	0.125	−0.162	−0.099
RGP	0.147	0.044	0.037	0.087	−0.042	−0.226	−0.254		−0.199	−0.163	−0.305 *	−0.067	−0.342 *	−0.353 *	−0.367 *
RGW	−0.013	−0.042	0.108	−0.120	0.173	0.091	0.122		0.048	−0.137	0.135	−0.046	0.260	0.031	0.190

^a^ DT, EAT, MaT, MiT, TD, CSR and R represent the daily temperature, effective accumulated temperature, maximum temperature, minimum temperature, temperature difference, solar radiation and cumulative solar radiation, respectively. ^b^ RSD-HD, RHD-MD, RHR, RL, RW, RCR, RCL, RGY, RP, RS, RGP and RGW represent relative SD-HD (SD-HD was the days from sowing date to heading date), relative HD-MD (HD-MD was the days from heading date to physiological maturity date), relative head rice rate, relative grain length, relative grain width, relative chalky grain rate, relative chalkiness level, relative grain yield, relative panicle m^−2^, relative spikelets panicle^−1^, relative grain filling percentage and relative 1000-grain weight, respectively. * denotes significant correlation at level 0.05 and ** denotes significant correlation at level 0.01 (in Pearson correlation).

## Data Availability

The data presented in this study are available on request from the corresponding author.

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
