# Peer review of "Effect of Temperature and Radiation on Indica Rice Yield and Quality in Middle Rice Cropping System"

_plants, 2022, doi:10.3390/plants11202697_

Round 1
Reviewer 1 Report
I have revised the article entitled “Effect of temperature and radiation on indica rice yield and quality under different climatic conditions”. Authors assessed the indica rice growth (six different cultivars, tolerant or sensitive to temperature) in different China regions and in two-yearly experiment.
Title
The title resulted very generic, and a more specific sentence could be more appropriate and useful in the future citation of the present manuscript.
Results
Line 73: specify the meaning of EAT because it is the first time that appears in the text.
Line 73-75: Both the sentences seem to be contradictory.
Table 1: it is not clear which statistical analysis was performed. One-way or two-way ANOVA?
After table 1, there are not line numbers.
Fig. 1 and Fig. 4 are present in the Discussion section. Fig 1 lacks statistical analysis.
Pag. 3, line 5: it is the first time that DT, MaT and MiT appear in the text: specify.
Pag. 3, line 6: it is not clear the kind of correlation was done, Pearson? Spearman? Why the authors say that the high temperature significantly influenced the rice growth duration if the correlation is negative?
Table 2: specify the kind of correlation.
Pag. 3, in the middle: which is RP? Maybe RGP?
Discussion
Authors reported that the rice growth differences are due to solar radiation and temperature but they did not report the solar radiation and temperature averages during both the years of the experiment and in the different regions of the experiment.
Pag. 7: the sentence “This result suggested we can also obtain high grain yield…” has to be supported by references.
Materials and Methods
It is not clear where the different regions are, China?
Specify which cultivars are sensitive and which are tolerant to temperature.
Specify how many plants authors used in the experiment in the different regions and of different cultivars. In results section, it is not clear that the results come from the mix of six cultivars.
Statistical analysis: which ANOVA was performed? Which correlation was performed?
I suggest the English revision by a mother tongue speaker (See line 84: difference is different?, paragraph 2.2 last line: similarly, there were also some differences….ect).
Author Response
Title
The title resulted very generic, and a more specific sentence could be more appropriate and useful in the future citation of the present manuscript.
Response: Thanks for your suggestions very much. We have modified “Effect of temperature and radiation on indica rice yield and quality under different climatic conditions” into “Effect of temperature and radiation on indica rice yield and quality in middle rice cropping system”. The modifications in the revised manuscript are in RED.
Results
Line 73: specify the meaning of EAT because it is the first time that appears in the text.
Response: Thanks for your suggestions very much. We have added it in the revised manuscript. The modifications in the revised manuscript are in RED.
Line 73-75: Both the sentences seem to be contradictory.
Response: Thanks for your suggestions very much. We have modified this two sentences into “Before the heading stage, the EAT showed slightly difference among different ecological regions except Yifeng (Table 1). The EAT of the six indica rice varieties in both years showed the same pattern under each area, except Yuyang.”
Table 1: it is not clear which statistical analysis was performed. One-way or two-way ANOVA?
Response: Thanks for your suggestions very much. We have added this in table 1. The modifications in the revised manuscript are in RED.
After table 1, there are not line numbers.
Response: Thanks for your suggestions very much. We have added line numbers in the revised manuscript.
Fig. 1 and Fig. 4 are present in the Discussion section. Fig 1 lacks statistical analysis.
Response: Thanks for your suggestions very much. We have modified line numbers in the revised manuscript. The growth durations were same among three repetitions for each variety. Therefore the growth duration were showed without repetitions. The Fig.1 were used to show the mean growth duration of six cultivars in different areas with ± standard error of the means of six cultivars (n=6).
Pag. 3, line 5: it is the first time that DT, MaT and MiT appear in the text: specify.
Response: Thanks for your suggestions very much. The modifications in the revised manuscript are in RED.
Pag. 3, line 6: it is not clear the kind of correlation was done, Pearson? Spearman? Why the authors say that the high temperature significantly influenced the rice growth duration if the correlation is negative?
Response: Thanks for your suggestions very much. Pearson correlation was conducted in this study. We have deleted “These results indicated that high temperature significantly influenced the rice growth duration, finally affected grain yield and grain quality” in the revised manuscript.
Table 2: specify the kind of correlation.
Response: Thanks for your suggestions very much. We have added this in table 2. The modifications in the revised manuscript are in RED.
Pag. 3, in the middle: which is RP? Maybe RGP?
Response: Thanks for your suggestions very much. RP means relative panicle m-2, RGP means relative grain filling percentage. We have added this in the revised manuscript. The modifications in the revised manuscript are in RED.
Discussion
Authors reported that the rice growth differences are due to solar radiation and temperature but they did not report the solar radiation and temperature averages during both the years of the experiment and in the different regions of the experiment.
Response: Thanks for your suggestions very much. To assess the correlation of climatic conditions with grain yield and quality, we evaluated the correlation of temperature and solar radiation with rice yield and quality. Like most cereals, rice growth season can be divided into different phenological stage, for example before heading stage and after heading stage. Previous studies suggested that It is effective to evaluate the effect of temperature at different phenological stages on rice yield and quality. Therefore, we studied the correlation of temperature and radiation during before heading stage and after heading stage in the present study and the temperature and radiation conditions before heading stage and after heading stage in Table 1 instead of showing the solar radiation and temperature averages during both the years of the experiment and in the different regions of the experiment.
Pag. 7: the sentence “This result suggested we can also obtain high grain yield…” has to be supported by references.
Response: Thanks for your suggestions very much. We have added the references in the revised manuscript.
Shi, W.J.; Yin, X.Y.; Struik, P.C.; Solis, C.; Xie, F.M.; Schmidt, R.C.; Huang, M.; Zou, Y.B.; Ye, C.R.; Jagadish, S. High day- and night-time temperatures affect grain growth dynamics in contrasting rice genotypes. J. Exp. Bot. 2017, 68, 5233–5245.
Materials and Methods
It is not clear where the different regions are, China?
Response: Thanks for your suggestions very much. We have added China in the revised manuscript.
Specify which cultivars are sensitive and which are tolerant to temperature.
Response: Thanks for your suggestions very much. We have provided data from each cultivar as supplemental table (Table S3), including grain yield, head rice rate, chalky grain rate and chalkiness level. The modifications in the support material are in RED. We can clearly understand 6 indica rice cultivars which were used in the experiment responded to climatic parameters in different manner clearly.
Specify how many plants authors used in the experiment in the different regions and of different cultivars. In results section, it is not clear that the results come from the mix of six cultivars.
Response: Thanks for your suggestions very much. Six cultivars were conducted in four areas with three repetitions. At the maturity stage, plants from an area of 5 m2 in the center of the plot were harvested. The 100 g subsample from each plot was used to determine rice quality. It was not the mix of six cultivars.
Statistical analysis: which ANOVA was performed? Which correlation was performed?
Response: Thanks for your suggestions very much. We have added it in table 1 and table 2 in the revised manuscript. One-way ANOVA and Pearson correltion were conducted in the present study. We also added this in Materials and Methods section.
I suggest the English revision by a mother tongue speaker (See line 84: difference is different?, paragraph 2.2 last line: similarly, there were also some differences….ect).
Response: Thanks for your suggestions very much. We have modified the language carefully. The grammar, sentence structure, and the language of our new manuscript have been carefully revised by an English native speaker.
Reviewer 2 Report
In this study, six cultivars field experiment was conducted to investigate the impact of climatic factors under four ecological regions with different indica rice cultivars, the main type of rice in east and central China. To explore how the different quality characteristics and grain yield are affected when grown under different ecological region, and to assess the correlation of climatic conditions with grain yield and quality. At present, innovation is not very great, but it has certain research significance. However, I think the following points of the article are unreasonable and need to be improved:1. The abstract does not match the title. There is no radiation content in the abstract.
2. In this study, six indica rice cultivars have different temperature sensitivities, so it is unreasonable to analysis the effect of temperature on rice yield and quality using the mean of six cultivars.
3. The order of the charts is chaotic, which seriously affects the readability of the article, pay attention to modifications.
4. The longer error bars in the figure indicate that the difference between the repetitions is large, and whether the difference between the repetitions will affect the difference between the treatments and lead to the impact on the result. It is recommended to verify the data or conduct data supplementation tests.
5. Data not shown in section 2.3 does not have to be described. The yield data of six varieties in different regions, correlations of grain yield and quality with humidity and light hours are not reflected, so it is not necessary to discuss.
6. Section 2.5 as a result of the analysis, the length is too long, it is recommended to compress and streamline.
7. Fig 1 and Fig 4 should follow the corresponding paragraph.
8. 3.1 In the first paragraph o, it is mentioned that the difference of growth duration is the reason for the difference of yield and quality in different planting areas and different years. Please add the result of how growth duration affects quality and cite relevant references.
9. The effects of radiation are less discussed in the discussion section, and the key effects of radiation on rice yield and quality are not specifically discussed. The title of this paper is “Effect of temperature and radiation on indica rice yield and quality under different climatic conditions”. However, the discussion and conclusion all mentioned that radiation has little correlation with rice yield and quality. It is suggested to delete radiation from the title and the paper should focus more on the effect of temperature on rice yield and quality.
10. The annual meteorological data of four site should be shown in 4.2, for example precipitation.
11. Correlation analysis was not enough performed to determine the effect of temperature and solar radiation on growth duration, grain yield and quality. It is a simple correlation coefficient, and it does not reflect the interaction between indicators.
12. Most of the discussion is describing the content of their own results, rather than concatenating the results of previous generations for discussion and analysis.
13. The blow molding machine in Section 4.3 needs to supplement the instrument information.
14. The format of the references is very irregular, please pay attention to the revision.
Author Response
RESPONSES TO EDITOR AND REVIEWER 2:
- The abstract does not match the title. There is no radiation content in the abstract.
Response: Thanks for your suggestions very much. This study demonstrated that temperature in these regions is a limiting factor compared with radiation. We have provided radiation content in the revised manuscript. The modifications in the revised manuscript are in RED.
- In this study, six indica rice cultivars have different temperature sensitivities, so it is unreasonable to analysis the effect of temperature on rice yield and quality using the mean of six cultivars.
Response: Thanks for your suggestions very much. The objectives of the study are to: 1) to explore how the different quality characteristics and grain yield are affected when grown under different ecological regions; 2) to assess the correlation of climatic conditions with grain yield and quality. For this purpose, six indica rice cultivars were grown under different ecological regions. In fact, the six indica rice cultivars have different temperature sensitivities. In order to identify the response of rice growth to temperature and radiation change, the differences in properties among cultivars were minimized by using the average and standardized management according to the method by Deng et al. 2015 and Deng et al. 2022. The standardized data were defined as “relative data” and and calculated as follows:
Relative data= (Observed data of a variety during at a specific study site in one years) / (Mean data of this variety across study sites in both years).
Deng, F.; Zhang, C.; He, L.; Liao, S.; Li, Q.; Li, B.; Zhu, S.; Gao, Y.; Tao, Y.; Zhou, W.; Lei, X.; Wang, L.; Hu, J.; Chen, Y.; Ren, W. Delayed sowing date improves the quality of mechanically transplanted rice by optimizing temperature conditions during growth season. Field Crops Res. 2022, 108493.
Deng, N.; Ling, X.; Sun, Y.; Zhang, C.; Fahad, S.; Peng, S.; Cui, K.; Nie, L.; Huang, J. Influence of temperature and solar radiation on grain yield and quality in irrigated rice system. Eur. J. Agron. 2015, 64, 37-46.
- The order of the charts is chaotic, which seriously affects the readability of the article, pay attention to modifications.
Response: Thanks for your suggestions very much. We have modified the order of the charts. The modifications in the revised manuscript are in RED.
- The longer error bars in the figure indicate that the difference between the repetitions is large, and whether the difference between the repetitions will affect the difference between the treatments and lead to the impact on the result. It is recommended to verify the data or conduct data supplementation tests.
Response: Thanks for your suggestions very much. In this study, the rice yield and grain quality showed similar trends across different cultivated areas and years, thus we use the means of six cultivars to show the differences across different cultivated areas and two years. The vertical bars represent standard error (± SE) of the means of six cultivars (n=6). It did not indicate the difference between the repetitions. We have verified the data carefully, the CV of rice growth duration, grain yield and quality among repetitions were range from 1.1% to 10.0%.
- Data not shown in section 2.3 does not have to be described. The yield data of six varieties in different regions, correlations of grain yield and quality with humidity and light hours are not reflected, so it is not necessary to discuss.
Response: Thanks for your suggestions very much. We have modified in the revised manuscript.
- Section 2.5 as a result of the analysis, the length is too long, it is recommended to compress and streamline.
Response: Thanks for your suggestions very much. We have modified in the revised manuscript. We have streamlined this section based on your suggestion. At the same time, according to the description of this section, we divided it into two parts so as to be more convenient for readers to understand.
- Fig 1 and Fig 4 should follow the corresponding paragraph.
Response: Thanks for your suggestions very much. We have modified in the revised manuscript. The modifications in the revised manuscript are in RED.
- 3.1 In the first paragraph, it is mentioned that the difference of growth duration is the reason for the difference of yield and quality in different planting areas and different years. Please add the result of how growth duration affects quality and cite relevant references.
Response: Thanks for your suggestions very much. We have added “Previous studies proved that rice growth duration determines the total amount of incident solar radiation, and high interception of solar radiation by the canopy is closely associated with high biomass production and grain yield (Katsura et al., 2008; Xu et al., 2018) Shortened grain filling duration would result in low head rice rate and high chalk grain rate (Huang et al., 2013)” in the revised manuscript.
Katsura, K.; Maeda, S.; Lubis, I.; Horie, T.; Cao, W.; Shiraiwa,T. The high yield of irrigated rice in yunnan, china: 'a cross-location analysis'. Field Crops Res. 2008, 107, 1-11.
Xu, L.; Zhan, X.; Yu, T.; Nie, L.; Huang, J.; Cui, K.; Wang, F.; Li, Y.; Peng, S. Yield performance of direct-seeded, double-season rice using varieties with short growth durations in central china. Field Crops Res. 2018, 227, 49-55.
Huang, M.; Jiang, L.; Zou, Y.; Zhang, W. On-farm assessment of effect of low temperature at seedling stage on early-season rice quality. Field Crops Res. 2013, 141, 63–68.
- The effects of radiation are less discussed in the discussion section, and the key effects of radiation on rice yield and quality are not specifically discussed. The title of this paper is “Effect of temperature and radiation on indica rice yield and quality under different climatic conditions”. However, the discussion and conclusion all mentioned that radiation has little correlation with rice yield and quality. It is suggested to delete radiation from the title and the paper should focus more on the effect of temperature on rice yield and quality.
Response: Thanks for your suggestions very much. Previous suggested that both temperature and radiation play key roles in determining grain yield as well as grain quality. Thus we have assessed the effect of temperature and radiation on rice development, grain yield and quality in the present study. However, compared with temperature, solar radiation showed slighter correlation with grain yield and quality. It was consistent with the finding that temperature is the most important climatic factor governing the growth duration of rice (Huang et al., 2013; Goswami et al., 2006 ). Moreover, solar radiation demonstrated strongly negative correlations with grain yield and quality in the present study, which went against the common sense of the positive influence of radiation on grain yield and quality (Wu et al., 2017; Deng et al., 2021). This result may be attributed to the fact that there were confounding effects of temperature and solar radiation (Deng et al., 2015). This means that high temperature always accompanied with high solar radiation in these regions, and the former have significantly negative effects on grain yield and quality (Shi et al., 2017; Tu et al., 2022). Similarly, previous studies have shown that the solar radiation is not a limiting factor to grain yield and quality, compared with temperature in these regions (Deng et al., 2015; Tu et al., 2020). Therefore, we suggested that the variations of grain yield and quality among different cultivated areas were mainly attributed to the temperature changes. We have these discuss in the revised manuscript. We also added “This study indicated that in these regions temperature is a limiting factor compared with radiation” in conclusion section.
Goswami, B.; Mahi, G.S.; Saikia, U.S. Effect of few important climatic factors on phenology, growth and yield of rice and wheat – a review. Agric. Rev. 2006, 27, 223–228.
Huang, M.; Jiang, L.; Zou, Y.; Zhang, W. On-farm assessment of effect of low temperature at seedling stage on early-season rice quality. Field Crops Res. 2013, 141, 63–68.
Wu, L.; Zhang, W.; Ding, Y.; Zhang, J.; Cambula, E.D.; Weng, F.; Liu, Z.; Ding, C.; Tang, S.; Chen, L.; Wang, S.; Li, G. Shading contributes to the reduction of stem mechanical strength by decreasing cell wall synthesis in japonica rice (Oryza sativa L.). Front. Plant Sci. 2017, 8, 881.
Deng, F.; Li, Q.P.; Chen, H.; Zeng, Y.L.; Li, B.; Zhong, X.Y.; Wang, L.; Ren, W.J. Relationship between chalkiness and the structural and thermal properties of rice starch after shading during grain-filling stage. Carbohydr. Polym. 2021, 252, 117212.
Shi, W.J.; Yin, X.Y.; Struik, P.C.; Solis, C.; Xie, F.M.; Schmidt, R.C.; Huang, M.; Zou, Y.B.; Ye, C.R.; Jagadish, S. High day- and night-time temperatures affect grain growth dynamics in contrasting rice genotypes. J. Exp. Bot. 2017, 68, 5233–5245.
Tu, D.; Jiang, Y.; Zhang, L.; Cai M.; Li, C.; Cao, C. Effect of various combinations of temperature during different phenological periods on indica rice yield and quality in Yangtze River Basin in China. J. Integr. Agr. 2021. Doi: 10.1016/S2095-3119(21)63803-0
- The annual meteorological data of four site should be shown in 4.2, for example precipitation.
Response: Thanks for your suggestions very much. We have shown the temperature and solar radiation condition in Table 1 during rice growth season. We add the light hours, humidity and precipitation in support material (Fig. S1).
- Correlation analysis was not enough performed to determine the effect of temperature and solar radiation on growth duration, grain yield and quality. It is a simple correlation coefficient, and it does not reflect the interaction between indicators.
Response: Thanks for your suggestions very much. Numerous studies have proved the effect of temperature and radiation on growth duration, grain yield and quality by controlling experiments. While, the objectives of the study are to: 1) to explore how the different quality characteristics and grain yield are affected when grown under different ecological regions; 2) to assess the correlation of climatic conditions with grain yield and quality. Thus, we first investigated the characteristics of temperature and radiation and the difference of rice growth duration, grain yield and quality across different cultivated areas and years. Then, we used correlation analysis to assess the correlation of climatic conditions with grain yield and quality according to the methods by Deng et al. (2015) and Deng et al. (2022). Result showed that growth duration, grain yield and quality were closely associated with temperature. It was consistent with the findings that high temperature would affect rice growth duration, yield and quality. In addition, according to previous studies findings that rice growth duration determines the total amount of incident solar radiation, and high interception of solar radiation by the rice canopy is closely associated with high biomass production and grain yield (Katsura et al., 2008; Xu et al., 2018). Shortened grain filling duration would result in low head rice rate and high chalky grain rate (Huang et al., 2013). Thus, we suggested that the grain yield and quality changed greatly due to the variations of growth duration among different cultivated areas even they planted the same cultivar.
Deng, F.; Zhang, C.; He, L.; Liao, S.; Li, Q.; Li, B.; Zhu, S.; Gao, Y.; Tao, Y.; Zhou, W.; Lei, X.; Wang, L.; Hu, J.; Chen, Y.; Ren, W. Delayed sowing date improves the quality of mechanically transplanted rice by optimizing temperature conditions during growth season. Field Crops Res. 2022, 108493.
Deng, N.; Ling, X.; Sun, Y.; Zhang, C.; Fahad, S.; Peng, S.; Cui, K.; Nie, L.; Huang, J. Influence of temperature and solar radiation on grain yield and quality in irrigated rice system. Eur. J. Agron. 2015, 64, 37-46.
Katsura, K.; Maeda, S.; Lubis, I.; Horie, T.; Cao, W.; Shiraiwa,T. The high yield of irrigated rice in yunnan, china: 'a cross-location analysis'. Field Crops Res. 2008, 107, 1-11.
Xu, L.; Zhan, X.; Yu, T.; Nie, L.; Huang, J.; Cui, K.; Wang, F.; Li, Y.; Peng, S. Yield performance of direct-seeded, double-season rice using varieties with short growth durations in central china. Field Crops Res. 2018, 227, 49-55.
Huang, M.; Jiang, L.; Zou, Y.; Zhang, W. On-farm assessment of effect of low temperature at seedling stage on early-season rice quality. Field Crops Res. 2013, 141, 63–68.
- Most of the discussion is describing the content of their own results, rather than concatenating the results of previous generations for discussion and analysis.
Response: Thanks for your suggestions very much. We have modified in the revised manuscript. The modifications in the revised manuscript are in RED.
- The blow molding machine in Section 4.3 needs to supplement the instrument information.
Response: Thanks for your suggestions very much. We add “ a seed blower (CFY-II, China) was used to separate half-filled and empty grains” in the revised manuscript. The modifications in the revised manuscript are in RED.
- The format of the references is very irregular, please pay attention to the revision.
Response: Thanks for your suggestions very much. We have modified in the revised manuscript. The modifications in the revised manuscript are in RED.
Reviewer 3 Report
In this manuscript, the authors reported that temperature and radiation in different climatic condition affect yield and quality of rice. The result is helpful for understanding environmental parameters’ effect on rice growth and metabolism, it is helpful for guiding rice cultivation as well. The data are sufficient to build a complete story. Meanwhile, there are still some points need to be addressed or provided.
1. The typesetting need to be adjusted. It is not fit to put Figure 1 and Figure 4 at the end of discussion.
2. Line 68, “….study are to:…”, “to” should be deleted.
3. Data in Figure 3 showed the relation between RGY with RP, RS and RGP etc. To my opinion, the results are not novel enough to be showed as a figure. It is a common sense that RGY is correlated with these parameters. Or, the authors need to discuss the necessity of showing these results.
4. The discussion is not concise enough. Some repetitive statements need to be deleted.
5. The authors mentioned that 6 indica rice cultivars which were used in the experiment responded to climatic parameters in different manner. I think the authors need to provide data from each cultivar as a supplemental figure or table. It may be helpful for readers to understand the story clearly.
Author Response
- The typesetting need to be adjusted. It is not fit to put Figure 1 and Figure 4 at the end of discussion.
Response: Thanks for your suggestions very much. We have modified in the revised manuscript. The modifications in the revised manuscript are in RED.
- Line 68, “….study are to:…”, “to” should be deleted.
Response: Thanks for your suggestions very much. We have modified in the revised manuscript. The modifications in the revised manuscript are in RED.
- Data in Figure 3 showed the relation between RGY with RP, RS and RGP etc. To my opinion, the results are not novel enough to be showed as a figure. It is a common sense that RGY is correlated with these parameters. Or, the authors need to discuss the necessity of showing these results.
Response: Thanks for your suggestions very much. Of course, rice yield is composed of panicles, spikelets per panicle, seed setting rate and 1000 grain weight. However, the grain yield not always showed closely correlation with all parameters. For example, grain yield was closely associated with spikelets in Xu et al. (2018), which indicated that the differences of grain yield among treatments were mainly induced by the differences of spikelets. Likewise, in the present study the grain yield showed greater correlation with panicles and grain filling percentage, which indicated that the difference of grain yield across cultivated areas and years were mainly associated with the variations of panicles and grain filling percentage. We have discussed this in discuss section in the revised manuscript.
Xu, L.; Zhan, X.; Yu, T.; Nie, L.; Huang, J.; Cui, K.; Wang, F.; Li, Y.; Peng, S. Yield performance of direct-seeded, double-season rice using varieties with short growth durations in central china. Field Crops Res. 2018, 227, 49-55.
- The discussion is not concise enough. Some repetitive statements need to be deleted.
Response: Thanks for your suggestions very much. We have modified in the revised manuscript. The modifications in the revised manuscript are in RED.
- The authors mentioned that 6 indica rice cultivars which were used in the experiment responded to climatic parameters in different manner. I think the authors need to provide data from each cultivar as a supplemental figure or table. It may be helpful for readers to understand the story clearly.
Response: Thanks for your suggestions very much. We have provided data from each cultivar as supplemental table (Table S3), including grain yield, head rice rate, chalky grain rate and chalkiness level. The modifications in the support material are in RED.
Round 2
Reviewer 1 Report
Authors improved the manuscript. I think that it could be accepted in the present form.
Reviewer 2 Report
The author had made many changes according to the opinion of the reviewer comments.